# Studying the Properties of Chromium-Contaminated Soil Solidified by Polyurethane

**DOI:** 10.3390/polym15092118

**Published:** 2023-04-28

**Authors:** Qiang Ma, Junjie Chen, Wentao Li, Nianze Wu

**Affiliations:** Innovation Demonstration Base of Ecological Environment Geotechnical and Ecological Restoration of Rivers and Lakes, School of Civil Engineering, Architecture and Environment, Hubei University of Technology, Wuhan 430068, China; chenghongliang@hbut.edu.cn (J.C.); wli20201027@hbut.edu.cn (W.L.); wunianze@hbut.edu.cn (N.W.)

**Keywords:** chromium-contaminated soil, polyurethane, unconfined compression test, solidified soil, toxicity characteristic leaching procedure

## Abstract

The solidification of chromium-contaminated soil using polyurethane (PU) was systematically investigated. The unconfined compression test was conducted to investigate the effects of the curing time, PU dosage and the content of chromium ions on the unconfined compressive strength (UCS) of chromium-contaminated soil. The effect of the PU dosage on the pore structure was investigated using nuclear magnetic resonance (NMR) and scanning electron microscopy (SEM), and the mechanism of strength change was revealed by combining the strength law with the pore structure development law. In addition, the ability of the PU to solidify the chromium-contaminated soil was studied by the toxicity characteristic leaching procedure (TCLP). According to the above test results, the UCS and the ability of the PU to solidify the chromium ions increased with the increase in curing time. The NMR tests showed that with the increase in PU dosage, the porosity decreased and the soil became more compact, hence increasing the strength. When the chromium ion content was 2000 mg/kg and the PU dosage was 8%, the strength of the sample was 0.37 MPa after curing for 24 h, which met the requirement of 0.35 MPa set by the U.S. Environmental Protection Agency. Consequently, PU is a solidification agent with high-early strength.

## 1. Introduction

Soil pollution in China has been very serious in the old heavy industrial areas, with the area of contaminated soils reaching 16.1% [1]. The United States, Japan, Germany, other developed countries and many developing countries have also faced the problem of metal-contaminated soil, and scholars in those countries have done a lot of work in the field of restoration [2,3,4,5]. With the continuous development of the chrome industry, chromium and its compounds are widely used in electroplating, printing, and photography, resulting in a large amount of chromium-containing industrial waste. Chromium is difficult to recycle, combined with improper landfilling practices, it thus produces a large amount of chromium-contaminated soil. According to the national soil pollution survey bulletin, the violation of the standard limit of chromium-contaminated soil has reached 1.1%. It has been estimated that the heavily contaminated soil in China has reached 12–15 million tons.

At present, the treatment methods for metal-contaminated soil mainly include solidification/stabilization (S/S), chemical remediation, electric remediation, phytoremediation, electrokinetic remediation and biological remediation [6]. From a practical and economic point of view, S/S is currently the most widespread and mature technology. In this technology, the binder is added to the contaminated soil, so that the binder and the soil are chemically solidified and physically adsorbed. The contaminant of heavy metals cannot be removed from the soil after solidification. Additionally, the commonly used binders are inorganic gel materials, such as cement and fly ash; organic gel materials, such as asphalt, polyester and polyethylene and chemical reagents, such as ferrous sulfate and phosphate [7,8]. Zhang [9] studied and confirmed that the chemical reagent iron sulfate has the ability to solidify the metal-contaminated soils. Xiong, et al. [10,11] found that inorganic gel materials, such as cement and nanofibrous can solidify chromium-contaminated soils and enhance their mechanical properties. PU is an organic gel material that is widely used in the construction industry with its adhesion, environmental protection and fast bonding speed. It is also a kind of polymer that is prepared by polyester polyols, polyether polyols or water, small molecular polyols, polyamines and diisocyanates or polyisocyanates through polymerization [12]. The main chain of PU can be divided into soft and hard segments [13]. Different proportions and the types of soft and hard segments in the PU can affect the performance of PU materials. According to the different types of soft segments, polyether PU and polyester PU could be divided [14]. Polyether PU has unique corrosion resistance, better moisture penetration characteristics, good adhesion and excellent hydrolysis resistance. When polyether PU is placed in a high humidity environment, it can be stable for a long time and has good antibacterial performance due to its excellent hydrolysis resistance [15]. To date, PU has been used to strengthen the railway roadbed and anchor slope with high strength and lightweight [16]. However, researchers have mainly focused their attention on the mechanical properties of PU rather than a novel binder to solidify the contaminated soil. Meanwhile, compared with common binders, such as cement, the curing time of PU is shorter and the PU will not cause secondary pollution also the PU is more environmentally friendly [17,18,19]. In addition, when compared with asphalt, PU production can be conducted at room temperature and does not need heat, thus it is more cost-effective [20]. To the best of our knowledge, this is the first work applying PU to solidify the chromium-contaminated soil.

Based on the above reasons, the leaching toxicity and strength of PU solidified chromium-contaminated soil were investigated in this study, and the ability of PU as a novel binder to solidify the chromium-contaminated soil was discussed. Moreover, the effect of the PU dosage on pore structure change in soil was investigated. In particular, the mechanism of strength change was revealed by combining the strength law with the pore structure development law in this paper.

## 2. Materials and Methods

### 2.1. Experimental Materials

The soil sample was collected from the campus of Hubei University of Technology, China. As shown in Figure 1, the test soil sample is clay. The buried depth of the clay was about 2 m. Before use, the raw clay was dried in a blast dryer at 105 °C and crushed. The crushed clay was sieved with a 2 mm standard sieve. The natural moisture concentration, limit moisture concentration and maximum dry density were determined according to “Chinese Highway Geotechnical Test Procedure (JTG3430-2020)”. The pH was determined according to “ASTM D4972-01”. The properties of the clay are given in Table 1.

The mineral composition of the test clay was measured using X-ray fluorescence spectroscopy (XRF) and the results are listed in Table 2. It may be seen that the main components of the test clay are SiO_2_, Al_2_O_3_ and Fe_2_O_3_. Note that the test specifications were according to the “Chinese General Rule of Wavelength Dispersion X-ray Fluorescence Spectrometry (JY/TO 16-1996)”.

As the PU has the function of cementation, compaction and encapsulation, the structure of the solidified clay is more compact. The moisture concentration should be increased to the optimal moisture concentration of the testing clay to achieve a better curing stability [21]. Therefore, when the chromium-contaminated soil was prepared manually, the optimal moisture concentration was slightly higher during the test. As a result, the preliminary test showed that the designed moisture concentration was 20%. The PU polyester purchased from Guangsheng Building Materials Co., Ltd, Shanghai, China. was used in the experiment, which was obtained by mixing the polymer polyol and polyisocyanate as shown in Figure 1.

### 2.2. Samples Preparation

At present, scholars mostly use the relevant saline solution to prepare contaminated clay samples [22,23,24]. Therefore, analytical pure chromium nitrate dichromate (CrN_3_O_9_·9H_2_O, the CAS number is 7789-02-8) was used to study the properties of chromium-contaminated soils in this paper. The preparation procedure of artificially contaminated clay was as follows:According to the national standards of metal pollution, the content of chromium ion was determined to be 1000, 2000 and 3000 mg/kg, which indicates the content of chromium-contaminated soil in China [25,26]. The mass of deionized water was calculated according to the moisture concentration. According to the relative atomic mass, certain weights of CrN_3_O_9_·9H_2_O were calculated (i.e., 1000, 2000, 3000 mg/kg). A certain mass of CrN_3_O_9_·9H_2_O and deionized water were put into an ion mixer. After stirring uniformly, they were poured into the beaker and sealed with a plastic wrap.The mass of the clay sample was calculated according to the size of the mold and the density of the clay sample. The prepared chromium nitrate dichromate solution and the crushed clay were mixed evenly. The mixed clay samples were sealed with fresh-keeping film and kept at 20 °C and 95% humidity for seven days. The moisture and chromium ions in the clay samples were fully transferred and the moisture became more uniform.A cylindrical mold with dimensions of 38.1 × 76.0 mm was used in this study. After curing, a certain amount of the PU was quickly added to the clay sample and fully stirred after static completion. Finally, the clay sample was pressed into the mold with a jack as shown in Figure 2. After the sample preparation, it was kept for a certain period of time at 20 °C and 95% humidity for subsequent experiments.

### 2.3. Test Method

#### 2.3.1. Unconfined Compression Experiment

This experiment refers to the standard of the “geotechnical test standard (GB/T 50123-2019)” and uses a microcomputer-controlled electronic universal testing machine of Naier Co., Ltd, Jinan, China. for testing. In the experiment, the axial strain rate was set to 1% per minute. Through pre-testing, the curing ages were set to 1, 3, 5, 7 and 14 days, which were more appropriate. The samples with curing ages of 1, 3, 5, 7 and 14 days were used to study the effect of the curing age on their strength. Additionally, the PU dosages of 2, 4 and 6% (the ratio of PU weight to dry clay weight) and chromium ion amount of 1000, 2000 and 3000 mg/kg were considered. In addition, the samples with the PU dosages of 0, 2, 4, 6 and 8% were used to study the effect of the PU dosage on the strength, while the chromium ion content was 2000 mg/kg and the curing ages were 1, 3, 5 and 7 days (see second part of Table 3). Furthermore, the samples with the chromium ion content of 1000, 2000 and 3000 mg/kg were prepared to study the influence of the clay with different chromium ion contents, while the PU dosages were 2, 4 and 6% and the curing age was seven days (see third part of Table 3).

#### 2.3.2. Toxicity Characteristic Leaching Procedure

According to the standard of the ASTM Method 1311-TCLP, the clay samples in the unconfined experiment were crushed, passed through a 9.5 mm sieve, dried at 50 °C and tested at pH ≥ 5. Glacial acetic acid was selected as a binder to prepare the samples according to the standard. The prepared samples were placed in a flip oscillation device. After 18 h of flip oscillation at a speed of 30 r/min, 15 mL of supernatant was extracted and stored in the refrigerator. After preparing all the samples, the leaching agent was placed in the refrigerator. The chromium ion content was measured using a SHIMADZU AA-6880 flame atomic absorption spectrometer. The sample details are listed in Table 4.

#### 2.3.3. Analysis of Soil Pore Distribution Change

The distribution of soil pores is an important factor affecting the strength of the soil. To investigate the effect of the PU on the soil pore distribution and to explain the mechanism of soil strength change, an NMR analyzer (Newmark Co., Ltd, Suzhou, China.) was used to study the change in soil pore distribution. The same method as described in Section 2.2 was used to prepare the artificial chromium-contaminated soil. The prepared contaminated soil was placed in a cylindrical mold with a diameter of 18 mm and a height of 30 mm. After curing for seven days and vacuum drying for 48 h, the samples were placed in a nuclear magnetic resonance device for analysis.

#### 2.3.4. Microstructural Analysis

The influence of the PU on the structure of the solidified soil can be directly observed by the morphology of the soil particles. In addition, the influence of the PU solidified chromium ions in the soil can be preliminarily judged by observing the binding morphology of the PU and the soil particles. A high-resolution field-emission scanning electron microscope was used to perform the SEM tests. Before use, the samples were placed in a freeze dryer, vacuum-dried for 48 h and then sprayed with gold. Finally, the samples were placed in the instrument for analysis. The images were magnified 1000 times.

## 3. Results

### 3.1. Toxicity Leaching Experiment

Figure 3 shows the TCLP experimental results. As shown in the figure, the leaching toxicity of PU-free samples does not change significantly with the curing times. The other four curves show a decreasing trend, which indicates that the amount of chromium ion leaching gradually decreases with the increase in the sample curing time. After seven days of curing, the chromium ion content is only 12.08 mg/kg, which is 70.07% lower than that of the sample after one day of curing. All the samples were lower than the risk control value of chromium ions (30 mg/kg), according to the Chinese standard GB 36600-2018. This is because there are a large number of polar groups on the PU molecule, which can effectively adsorb the chromium ions as a polar substance. As the sample curing time increases, the leaching amount of the chromium ions gradually decreases. This confirms studies on the solidification ability of the PU [27]. In addition, when the curing times are 1, 3 and 5 days, the effect of the PU dosage on the chromium ion content is not obvious. From the diagram, it can be seen that the chromium ion content of the sample decreases sharply when the curing time is 5–7 days. This is because the curing time is insufficient and the solidification effect is not obvious. After seven days of curing, the chromium ion content decreases when increasing the PU dosage. The TCLP experiments show that the PU can effectively solidify chromium-contaminated soil, and the solidification ability gradually increases with the increase in curing time.

### 3.2. Change in Sample Strength with Curing Time

Figure 4 shows the effect of the curing time on the strength of samples under three chromium ion contents. According to the pre-test, the curing ages are 1, 3, 5, 7 and 14 days. By fitting a curve with experimental data, the following function is suggested:(1)$$y=y_{0}-A_{1}e^{-x/t}$$ where *y_0_* is the PU dosage parameter that increases with the increase in the PU dosage, *A*_1_ is a correction factor, and *t* is the time parameter. Note that the larger the *t* is, the slower the function growth. The *R^2^* correlation coefficients of the curves under the three chromium ion contents are greater than 0.9, and therefore the correlation is good.

From Figure 4, it can be seen that under the three different contents of chromium ions, all the strengths increase with the increasing curing time. The three functions are arranged from top to bottom with the increase in the PU dosage. This indicates that under the three chromium ion contents, the strength of the sample improves significantly with the increase in the curing time. Under the condition of the same curing time, the higher the PU dosage, the higher the strength of the sample. When the chromium ion content is 1000 mg/kg, the strength of samples with the PU dosages of 2, 4 and 6% reach 0.359, 0.405 and 0.497 MPa after 14 days of curing. It also shows an increase in 215.3, 221.3 and 132.7% when compared with one day of curing. In addition, as the curing time increases, the curves tend to flatten gradually. This indicates that the increase in strength gradually slows down as the curing time increases. The reason for this is that it takes some time for the PU to solidify. During the solidification, the volume of the PU expands and fills the internal pores of the clay without increasing the overall volume of the clay, resulting in a denser clay. Hence, the strength of the clay increases significantly.

Figure 5 shows the increase in strength over time. The numbers 1, 2 and 3 correspond to the samples with the PU dosages of 2, 4 and 6 %, respectively, (note that for all cases, the chromium ion content is 1000 mg/kg). From Figure 5, we can see that the UCS of the three groups of experiments increases slowly in 7–14 days, while less than 10% of the total strength increases in 14 days. This indicates that the chromium-contaminated soil solidified with the PU can achieve high strength in a short period of time.

### 3.3. Change in Sample Strength with Dosage of PU

From Figure 6, we can see that when the content of the chromium ion is 2000 mg/kg and the curing times are 1, 3, 5 and 7 days, the curves have an increasing trend. Therefore, it indicates that the strength of the samples increase when increasing the PU dosage. When the dosage of the PU is 8% and the curing times are 1, 3, 5 and 7 days, the UCS reaches 0.372, 0.412, 0.459 and 0.469 MPa, respectively. With the curing time of 1, 3, 5 and 7 days, the strength of the four samples increases by 200, 227.3, 242.9 and 247.2%. The UCS of the sample cured in one day is allowed to reach 0.35 MPa as specified by the US Environmental Protection Agency [28]. When compared with chromium-contaminated soil solidified with cement [11], the curing time of the chromium-contaminated soil solidified with the PU is greatly reduced, while it has the characteristics of a high performance and high strength. There are two main reasons for the increase in strength. First, the PU has a strong bonding effect. The clay forms a quasi-cohesion, further strengthening the cohesion of the clay, thereby enhancing the strength of the clay. Some research [29,30] shows that this quasi-cohesion increases with the increase in the PU dosage. Hence, with an increase in the PU dosage, the UCS of the sample increases. Second, the volume of PU increases during solidification. The PU fills the pores between the clay particles and forms an irregular chain or discontinuous body. Hence, the PU can reduce the porosity of the clay to increase the strength of the clay. With the increase in the PU dosage, the pores in the clay are continuously filled. Therefore, with the increase in PU dosage, the UCS of the clay samples increases continuously.

In addition, under the condition of four kinds of curing times, it can be seen that the curves of 0–4% and 6–8% are relatively flat. Moreover, the curve of 4–6% is relatively steep. This indicates that in the process of 0–4% and 6–8%, the strength of the samples improves slowly. However, the strength of the sample improves very quickly in the process of 4–6%. As shown in Figure 7, the sample in the process of 4–6% has the highest percentage increase in strength of the four considered curing time conditions. In the process of 4–6%, the strength of the sample increases by 0.084, 0.087, 0.116 and 0.103 MPa under the four curing time conditions, which has the maximum growth rate. However, with the increase in the PU dosage, the increase in strength slows down. This is because the volume expansion of the sample is very high when the PU dosage is 8%. The expansion of the PU not only fills the pores of the clay, but also increases the volume of the clay. This may weaken the strength of the sample or lead to a slow growth in strength. When the dosage of PU is 6%, the curing time is three days, and the strength of the sample can meet the requirement of 0.35 MPa specified by the US EPA. Hence, taking both the engineering cost and the mechanical properties into consideration, the PU dosage of 6% is selected as the optimal dosage for the solidification of chromium-contaminated soil.

### 3.4. Variation in Strength with Chromium Ion Concentration

Figure 8 shows the effect of the different contents of chromium ions on strength. The UCS of the samples was tested when the curing time was seven days and the PU dosages were 2, 4 and 6%. As shown in the figure, the three functions are arranged from top to bottom with the increase in chromium ion concentration. This shows that under the same PU dosage, the strength of the sample decreases with the increase in chromium ion concentration. This is because after the addition of chromium ions in the clay, the free chromium ions are positively charged. In addition, the size of the clay particles is very small and the specific surface area is large. It is easy to absorb the positively charged chromium ions to form new particles, so the clay concentration in the sample is significantly reduced. This leads to the weakening of the cementation strength between the clay particles and the reduction in the stability of the clay structure and, as a result, the reduction inf soil strength [31,32]. With the increase in the PU dosage, the strength decreases gradually. When the PU dosage is 4%, the strength is reduced to 0.435 MPa and 0.175 MPa. On the other hand, when the PU dosage is 6%, the strength is reduced to 0.315 MPa and 0.110 MPa. This is because there are a large number of polar groups on the PU molecule. These polar groups form van der Waals forces and hydrogen bonding are force to adsorb heavy metal ions, which are the same polar substances. Some of the chromium ions are wrapped by the PU to reduce the contact between the clay particles and the chromium ions in the soil. Therefore, with the increase in the PU dosage, the strength decreases gradually. The decrease in strength gradually slows down. Zhang [29] found that when the chromium ion content was 10,000 mg/kg, the UCS of clay decreased by 53%, which is consistent with the results of this study.

### 3.5. The Effect of PU on Sample Pores

Figure 9 is the result of the NMR test, which shows the effect of the different PU dosages on porosity. The spherical pore model is used for an approximate simulation. Through the systematic inversion tool, the relaxation time and the NMR signal strength are transformed into the aperture radius (X axis) and the aperture ratio (Y axis), respectively. As shown in the figure, under the three dosages of the PU, the pore size is distributed in a bimodal pattern. According to the division law of pores in the existing silty clay [6,33], the pores were divided into micropores (r ≤ 5 µm), mesopores (5 µm ≤ r ≤ 20 µm) and macropores (r ≥ 20 µm) based on the equivalent diameter of the clay pores. The main peak is micropores, and the secondary peak is composed of micropores and mesopores. With the increase in the PU dosage, the main peak height gradually decreases and the peak spectrum slightly shifts to the left. At the concentration of 2, 4 and 6%, the pore size ratio 0 ≤ r ≤ 1 µm is 19.84, 18.95 and 18.57%. At the concentration of 2%, 0.036% of the pore size is 1 µm ≤ r ≤ 1.6 µm, and the pore size has a decreasing trend. This indicates that the quasi-cohesive force generated in the process of the PU curing makes the clay more closely bound. The quasi-cohesive force generated at low PU dosage is limited. With the increase in the PU dosage, the quasi-cohesive force in the clay gradually increases, and the various particles in the clay are cemented together. The irregular continuum is formed and the clay becomes denser. The peak spectra of 2, 4 and 6% are included in turn. As a whole, the area of the peak spectra decreases gradually with the increase in the PU dosage. The porosity is 20.18, 19.13 and 18.91%. This indicates that the volume of the PU expands during the curing, and the pores are continuously filled by the PU. Although the PU foam can continuously produce fine pores, the final result shows that the porosity decreases gradually. This indicates that the positive effect of the PU-filled pores is greater than the negative effect of its pores. With the increase in the PU dosage, the porosity decreases, the clay becomes denser and the clay strength increases. This phenomenon also explains the conjecture of 3.3.

### 3.6. Results of Scanning Electron Microscope

Figure 10 shows the solidification mechanism of chromium ions by the PU. Figure 10a corresponds to the sample without the PU. On the other hand, Figure 10b is related to the sample with the PU dosage of 6%. From Figure 10b, it can be seen that there are two main ways for the PU to affect the chromium-contaminated soil. One is wrapping and covering the clay particles, forming a closed film to seal the clay, reducing the transfer of heavy metals and effectively solidifying the chromium-contaminated soil. The other is bridging and filling, filling the gap between the two soil particles and bridging between the two soil particles. It is through these two ways that the PU reduces the porosity of the soil, greatly enhances the strength of the soil and reduces the precipitation and transfer of heavy metals.

## 4. Discussion

According to the unconfined compressive test, it was found that the UCS of the PU solidified chromium-contaminated soil is positively correlated with the PU dosage and the curing time. The chromium ions reduced the UCS of the sample. This result is probably due to the following reasons: First, the PU can fill and bridge the gap between the two soil particles. The quasi-cohesive force generated companying with PU filling and bridging. Moreover, the volume of the PU expands during the curing process, and the pores are continuously filled by the PU. With the increase in the PU dosage, the porosity decreases, the clay becomes denser and the UCS of the clay increases. The PU solidified chromium-contaminated soil can reach a high strength in a short period of time. The UCS of the samples increased slowly in 7–14 days, less than the 10% of the total strength increased in 14 days. This indicates that the PU is a solidification agent with high-early strength.

The TCLP shows that the PU can effectively solidify chromium-contaminated soil, and the solidification ability gradually increases with the increase in curing time and PU dosage. Some scholars have studied the adsorption mechanism of the PU chromium ions in polluted water, but the solidification mechanism of the PU chromium ions in soil is still unclear. This is an important topic for future research.

## 5. Conclusions

Metal-contaminated soil in the old industrial bases of China is very serious. Moreover, common binders have weaknesses, such as high costs and environmental pollution. To solve the above problems, the feasibility of a PU as a novel binder to solidify chromium-contaminated soil was discussed, and the leaching toxicity and mechanical properties of PU solidified chromium-contaminated soil were investigated in this study. Based on the unconfined compression test, the toxicity characteristic leaching procedure and the micro–tests at different PU dosages, the main conclusions are as follows:PU can effectively solidify chromium-contaminated soil in a short period of time. After curing for seven days, the chromium ion leaching value in all the samples was lower than the risk control value of the chromium ions.PU is a solidification agent with high-early strength. With the increase in curing time and PU dosage, the UCS was significantly improved. The sample with 8% PU could reached 0.35 MPa as specified by the United States Environmental Protection Agency after one day of curing.PU increasing the strength of the chromium-contaminated soil by decreasing the porosity of the soil. With the increase in the PU dosage, the number of micropores decreased significantly so the soil became denser and the strength of the soil was improved. The SEM test showed that the mechanism of the PU to solidify the chromium- contaminated soil might be as follows: adsorbing heavy metal ions and wrapping the soil particles to reduce the transfer ability of the chromium ions.This study implied that the PU, as a novel binder, can effectively solidify the chromium-contaminated soil. Taking both the engineering cost and mechanical properties into consideration, the PU dosage of 6% was chosen as the optimal dosage for the solidification of chromium-contaminated soils.

## Figures and Tables

**Figure 1 polymers-15-02118-f001:**
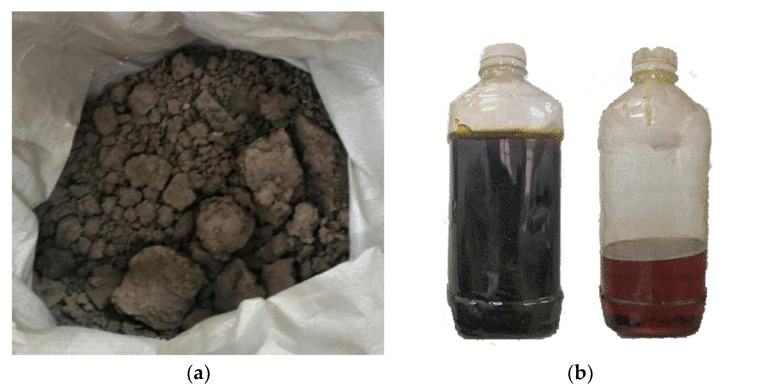
Materials of the sample (**a**) tested clay, (**b**) polyester PU (left is polymer polyol, right is polyisocyanate).

**Figure 2 polymers-15-02118-f002:**
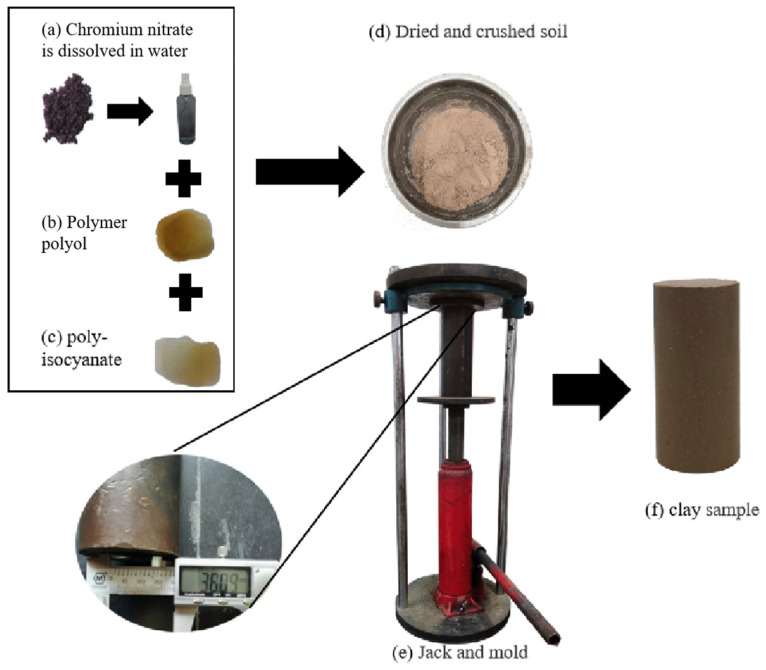
Preparation of sample materials and mold.

**Figure 3 polymers-15-02118-f003:**
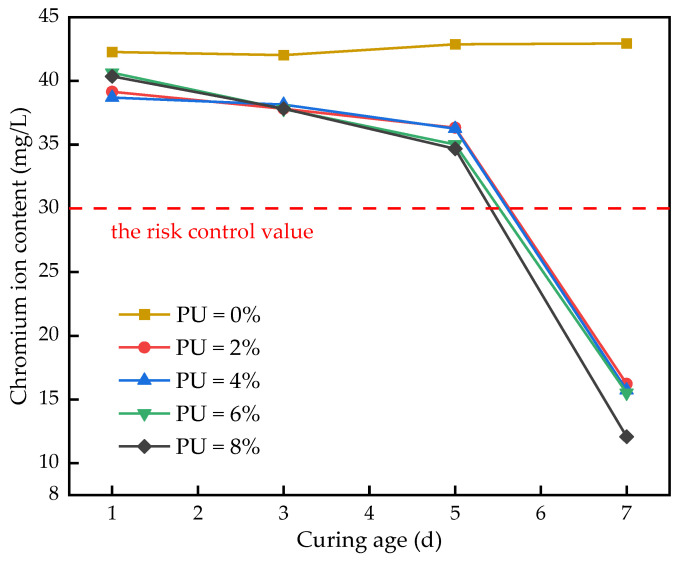
Leaching content of chromium ion.

**Figure 4 polymers-15-02118-f004:**
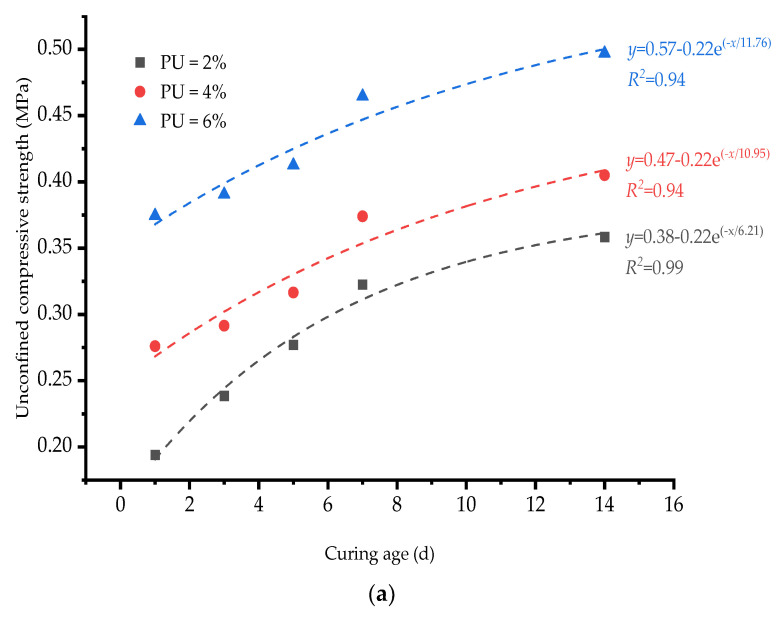
Variation in unconfined compressive strength (UCS) with curing time for different chromium ion contents: (**a**) 1000 mg/kg, (**b**) 2000 mg/kg and (**c**) 3000 mg/kg.

**Figure 5 polymers-15-02118-f005:**
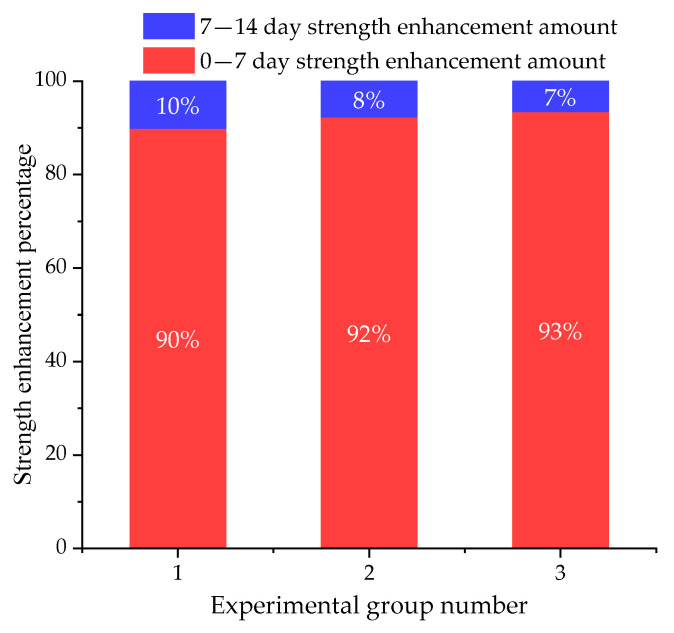
Sample strength changes with increasing solidification time.

**Figure 6 polymers-15-02118-f006:**
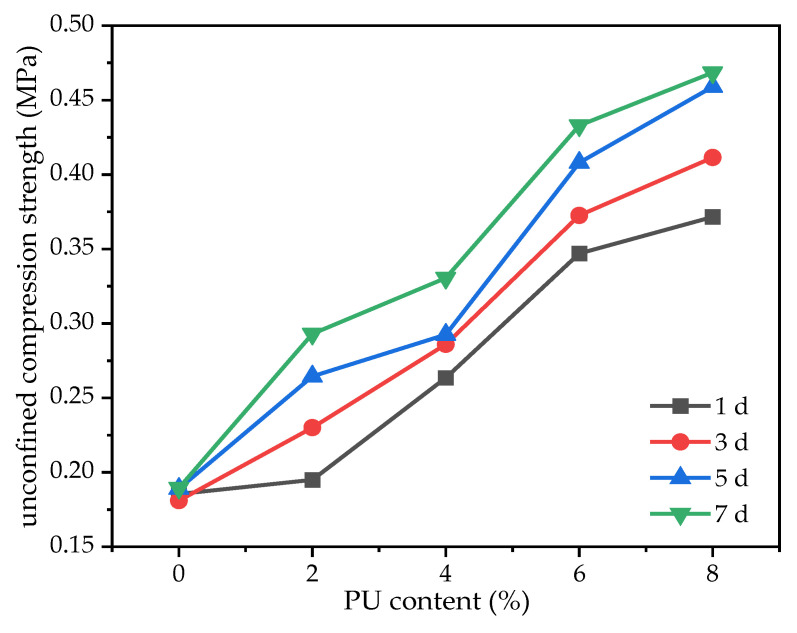
Variation in sample strength with PU dosage (2000 mg/kg).

**Figure 7 polymers-15-02118-f007:**
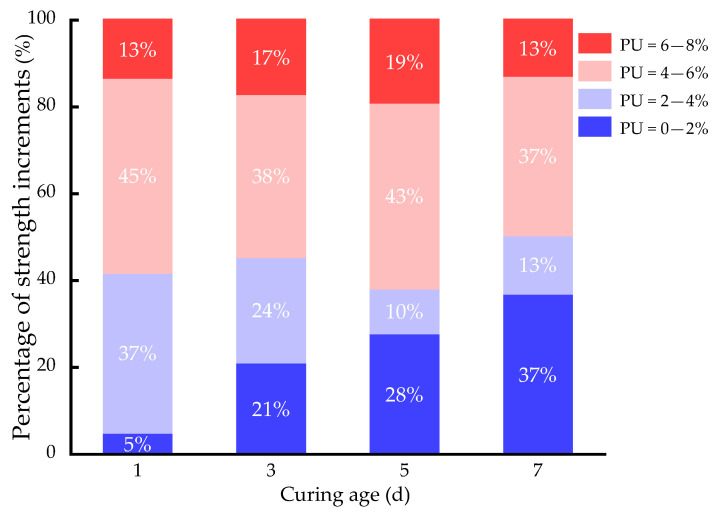
Percentage improvement in strength of samples with different dosages.

**Figure 8 polymers-15-02118-f008:**
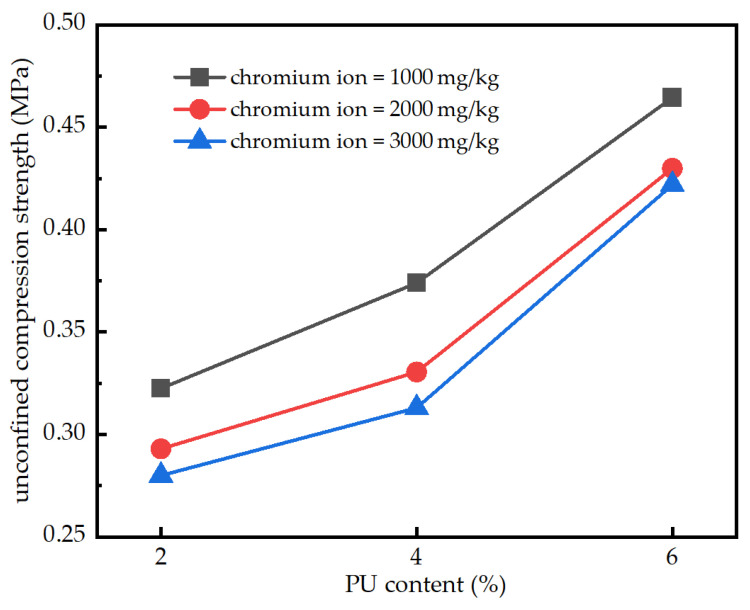
The influence of different chromium ion concentrations on the UCS of the sample.

**Figure 9 polymers-15-02118-f009:**
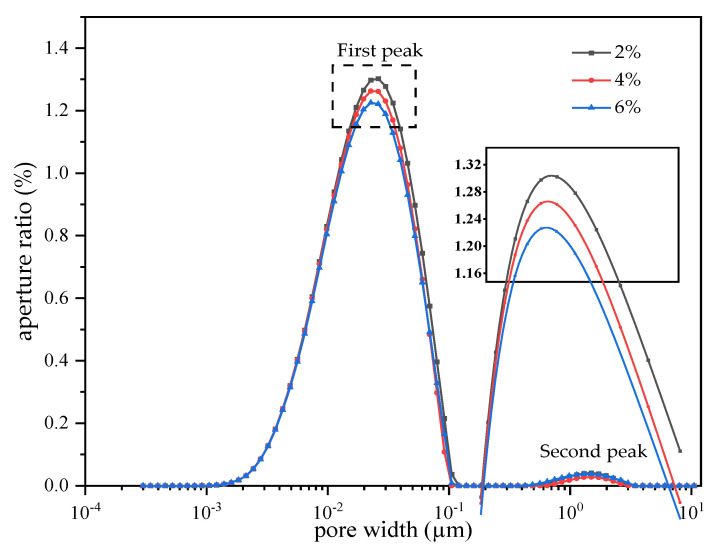
Effect of different PU dosages on porosity.

**Figure 10 polymers-15-02118-f010:**
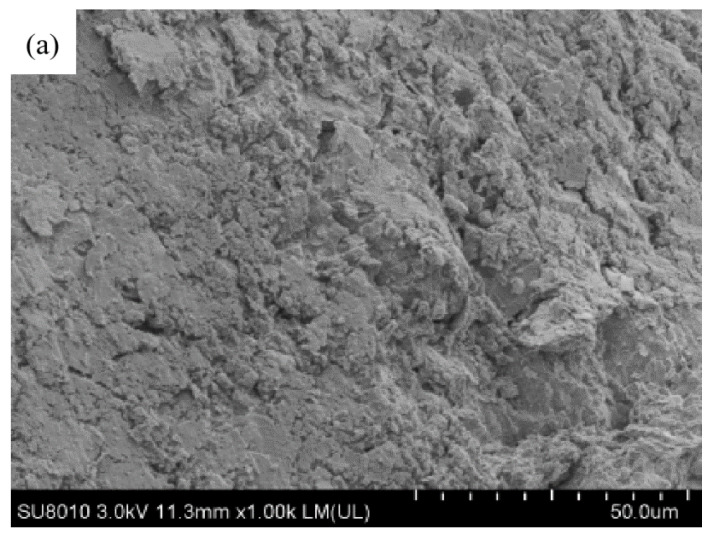
Results of scanning electron microscope experiment. (**a**) sample without PU, (**b**) sample with 6% PU.

**Table 1 polymers-15-02118-t001:** Main properties of uncontaminated clay samples used in the experiment.

Properties	Value
Dry density (g/cm^3^)	1.78
Specific gravity (g/cm^3^)	2.69
Optimal moisture concentration (%)	19.5
Natural moisture concentration (%)	32.00
Liquid limit (%)	48.20
Plastic limit (%)	20.10
pH	7.43

**Table 2 polymers-15-02118-t002:** Mineral composition of tested clay.

Element	SiO_2_	MgO	CaO	Al_2_O_3_	Fe_2_O_3_	K_2_O	Na_2_O	TiO_2_
Content (%)	63.90	2.19	3.66	18.69	6.45	2.59	0.74	0.95

**Table 3 polymers-15-02118-t003:** Samples in unconfined compression experiment.

Research Variable	Properties					
**Curing age**	**PU dosage**	**chromium ion content 1000 mg/kg,** **2000 mg/kg, 3000 mg/kg**				
		**Curing age**				
	2%	1 d	3 d	5 d	7 d	14 d
	4%	1 d	3 d	5 d	7 d	14 d
	6%	1 d	3 d	5 d	7 d	14 d
**PU dosage**	**PU dosage**	**chromium ion content 2000 mg/kg**				
		**Curing age**				
	0%	1 d		3 d	5 d	7 d
	2%	1 d		3 d	5 d	7 d
	4%	1 d		3 d	5 d	7 d
	6%	1 d		3 d	5 d	7 d
	8%	1 d		3 d	5 d	7 d
**chromium ion concentration**	**PU dosage**	**Curing 7 d**				
		**chromium ion concentration**				
	2%	1000 mg/kg	2000 mg/kg		3000 mg/kg	
	4%	1000 mg/kg	2000 mg/kg		3000 mg/kg	
	6%	1000 mg/kg	2000 mg/kg		3000 mg/kg	

d = day.

**Table 4 polymers-15-02118-t004:** TCLP samples setting chromium ion 2000 mg/kg.

PU Concentration	Curing Age			
0%	1 d	3 d	5 d	7 d
2%	1 d	3 d	5 d	7 d
4%	1 d	3 d	5 d	7 d
6%	1 d	3 d	5 d	7 d
8%	1 d	3 d	5 d	7 d

d = day.

## Data Availability

All data generated during this study are included in this published article. All data in the submitted work are either original work created by the authors listed on the manuscript or work for which permission to re-use has been obtained from the creator.

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
