# Peer review of "Studying the Properties of Chromium-Contaminated Soil Solidified by Polyurethane"

_polymers, 2023, doi:10.3390/polym15092118_

Round 1

Reviewer 1 Report

This work is very interesting research. The only regret is the too many grammatical errors in the document. They are so many that I prefer to include the original PDF file with yellow highlighted text that for sure are not the unique and could be more changes that going to improve the presentation of your work.

The English writing style is good enough but the punctuation marks have errors that seem terrible.

Reviewer 2 Report

The manuscript is about solidifying contaminated soil to increase its use and decrease the risks.  Soil solidification of polluted soil is done normally by cement, making the contaminated soil suitable to build a high landfill with a low permeability. When using proper covers, such landfills can be controlled well against leaching for a very long time. In this case the use of the solidification is not defined: roads, landfill, parks? The type if use is very relevant for the leaching in the future. Another important use of solidification is its use for sludges, especially after soil cleaning. The sludges cannot be used in a landfill if they are not solidified.

In the current manuscript it is shown that you can solidify the soil, an effect of chromium on the solidification is shown, and the authors state that an effect of the solidification on the leaching is shown.   

1.     No data are shown for the leaching of the blank soil (without PU). How can the authors state that solidification has an effect on the leaching of Cr when they show no data? On might argue that the authors show an effect of adding various amounts of PU, but figure 4 shows no clear differences between 2, 4, 6 and 8% PU. The effect of curing age might also happen for a sample with the blank (0% PU). So, the manuscript misses proof that PU has an effect on Cr leaching!

2.     The chromium used is not clear: the authors state that is chromium nitrate dichromate, but the chemical notation (CrN3O9 9H20) suggests only Cr3+.  

3.     The problem of chromium pollution in China and other countries is in most cases Cr3+ (trivalent chromium), so it important to be very clear if there is a problem. Chromate (hexavalent chromium) is very toxic. Cr3+  (trivalent chromium) is mostly not toxic, and is very stable in soils if organic matter is present, and there is no need to show an effect of solidification. In many cases the polluted soils not only contains Cr but also other heavy metals, so why focus on Cr? If PU has an effect op Cr it does not necessarily have the same effect on other pollutants? Why have the authors not used a real polluted soil?  

4.     The authors have not compared their technique with a mor conventional techniques such a solidification, for example cement. Without a comparison one cannot judge if the new technique is an improvement.

The use of English is not well.

Minor comments

29 “rate”. An area of 16.1% is not a rate.

30 “had”. Have

31 “had”. Have

33 “were” are

34 “was” is

Check the use of the English language in the rest of the text.

36 “the exceeding standard”. The percentage of the area exceeding the standard..

66 “more environmental friendly”. Interesting, but this is not obvious, so please explain.

102 “CrN3O9” unclear. Please give also other notations, and CAS number.

145 Showing a picture of an flame atomic absorption meter is awkward. It is a normal well known machine. Showing a picture of a shaking or turning machine is also awkward as it is part of any laboratory.

176-177 I cannot see that a increase of PU dosage results in lower amount of leaching. Without clear effects, repetitions, or statistical analysis it is difficult to make such statements.

see above

Round 2

Reviewer 2 Report

The authors have improved their paper. Very nice that you have been able to add the measurements of the Cr leaching without solidification. That makes the story much better. Strange that you did not give these data in the first admission. Please check Table 4, add a line with 0% PU.

Please remove the photos' of the AAS and the rotating machine! Nobody presents these standard lab materials.

Please check: in figure 6 (new number) you give the compression strength also for 0% PU. Why do you not give these measurements also in figure 4 and 8?

The use of English has been improved.
